# Conditionally Adaptive Graph Attention Networks for Credit Card Fraud Detection

## Abstract

Fraudulent transactions have been on the rise, leading to significant financial losses annually. In credit card fraud detection (CCFD), various predictive models aim to mitigate these losses by assessing transaction risk. While GNN-based methods have been employed to capture spatio-temporal transaction features, they often suffer from oversmoothing as graph layers increase, causing fraudulent and legitimate transactions to become indistinguishable. Existing semi-supervised methods that mask some labels have not fully resolved this issue. To address this, we propose the Multi-head Attention Conditional Variational Autoencoder (Ma-CVAE), which leverages weight distributions from imbalanced datasets and the Gumbel softmax distribution to construct more diverse reconstructed features, reducing feature homogenization. Then, we utilize Temporal Graph Attention Networks (TGAT) with a Multi-Attention mechanism to model risk propagation among transactions. Finally, classification probabilities are mapped to risk scores via a Multi-Layer Perceptron (MLP). Our approach achieves state-of-the-art performance, improving AUC scores by 1.45%, 3.05%, and 0.83% on three semi-supervised datasets: FFSD, YelpChi, and Amazon, respectively.

## 1 Introduction

The rise of online commerce has transformed commodity trading, with over 2.28 billion credit cards issued in a single quarter Bin Sulaiman et al. (2022). Credit cards have become a widely preferred payment method Wu et al. (2019); Zhu et al. (2023), significantly boosting transaction efficiency. Consequently, many predictive models have been developed to detect fraud and generate risk scores, functioning as binary classifiers. These models assist financial experts in prioritizing high-risk transactions and refining model performance.

Recently, deep learning models such as LSTM Jurgovsky et al. (2018), CNN Chen & Lai (2021); Muppalaneni et al. (2019), Transformer Benchaji et al. (2021), and generative models such as GAN Fiore et al. (2019); Ibrahim et al. (2020) have gained traction in CCFD tasks. However, these methods often overlook relational dependencies between transactions, leading to a disjointed learning process for different label types. This limitation underscores the need for models that can effectively capture these dependencies and improve the accuracy of fraud detection.

To address this, GNN-based models have been proposed to leverage relational information, showing promising results Zhang et al. (2020); Qiao et al. (2023); Liu et al. (2021a); You et al. (2024). Deeper layers in GNNs have the potential to capture complex and subtle features more effectively, as they allow the model to aggregate information from a larger neighborhood, leading to a richer understanding of the underlying data structure. However, with increasing depth, these models often suffer from over-smoothing, where node representations become overly similar, resulting in feature homogeneity. This causes performance to peak at a certain shallow depth before degrading with further layer stacking Rusch et al. (2023); Giraldo et al. (2023); Liang et al. (2024); Shen et al. (2024). Even state-of-the-art (SOTA) models like TGAT Xiang et al. (2023) are not immune to this issue, as the aggregation process diminishes the distinction between fraudulent and legitimate features when excessive layers are used. To qualitatively assess feature diversity and differentiation, we use the clustering method suggested by Tang et al. (2024a;b). As shown in Figure **??**(a), the original features are projected into a 2D space using PCA and clustered with DBSCAN Ester et al. (1996); Bai et al. (2021). The lack of clear cluster separation in Figure (a) indicates high feature

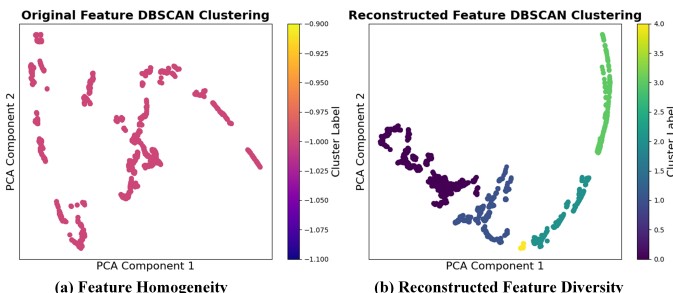

Figure 1: Illustration of diversity enhancement after Ma-CVAE processing. Input features are projected into a 2D space using PCA for visualization, followed by DBSCAN to identify clusters in dense regions. Under identical parameters, the unprocessed features in Figure 1 (a) form only one cluster. This shows that Ma-CVAE enhances feature diversity as illustrated in Figure 1 (b), leading to greater dispersion and variation.

similarity, highlighting the homogeneity issue that potentially leads to a loss of distinction between fraudulent and legitimate features.

Therefore, this paper proposes a Multi-head Attention-Conditional Variational Autoencoder (Ma-CVAE) model that integrates numerical, categorical, and transaction-related features. These features are mapped into the Gumbel-Softmax space Jang et al. (2017) to enhance diversity and simulate evolving fraud strategies, mitigating over-smoothing and over-squashing issues observed in GNN-based models Xiang et al. (2023); Dong et al. (2024); Cheng et al. (2020); Liu et al. (2020). As shown in Figure 1, features before and after Ma-CVAE processing demonstrate a higher number of clusters with the same number of features, indicating greater feature differentiation and diversity, thereby alleviating the issue of feature homogeneity as the number of feature layers increases. To handle the challenge of large volumes of unlabeled real-time data Xiang et al. (2023); Wang et al. (2019b), a semi-supervised dataset is simulated, with some feature labels randomly masked. The Ma-CVAE reconstructs these features, which are then passed through a Gated Temporal Attention Network Xiang et al. (2023) with a multi-head attention to capture spatial-temporal patterns and focuses on fraudulent transactions and their surrounding contexts. Finally, a two-layer MLP generates a transaction risk score based on these processed features. The contributions of this paper are outlined as follows:

- Our model enhances feature diversity during training and improves sensitivity to variations by mapping features into the Gumbel-Softmax space, effectively mitigating over-smoothing issues in graph-based methods.

- By applying a multi-head attention mechanism to label information, the method better handles label distributions, maximizing the benefits of semi-supervised learning.

- The model surpasses state-of-the-art methods across multiple metrics on the FFSD, Amazon, and YelpChi datasets, demonstrating superior performance in fraud detection tasks.

## 2 RELATED WORK

Various machine learning techniques have been explored to address credit card fraud detection. Early attempts Maes et al. (2002) involved methods such as Bayesian belief networks (BBN) and artificial neural networks (ANN), with ANN showing superior performance in real-world datasets. In subsequent studies, neural networks consistently outperformed decision trees Sahin & Duman (2011) in fraud detection tasks. Convolutional models Chen & Lai (2021); Muppalaneni et al. (2019) also demonstrated improved accuracy by capturing spatial patterns more effectively than traditional methods like SVM Sahin & Duman (2011), Random Forest Xuan et al. (2018), and XGBoost Trisanto et al. (2021); Ileberi et al. (2021). Additionally, ensemble techniques such as AdaBoost Freund & Schapire (1997) and majority voting were employed to further enhance detection accuracy. More recent work proposed an improved LSTM Jurgovsky et al. (2018) model to better

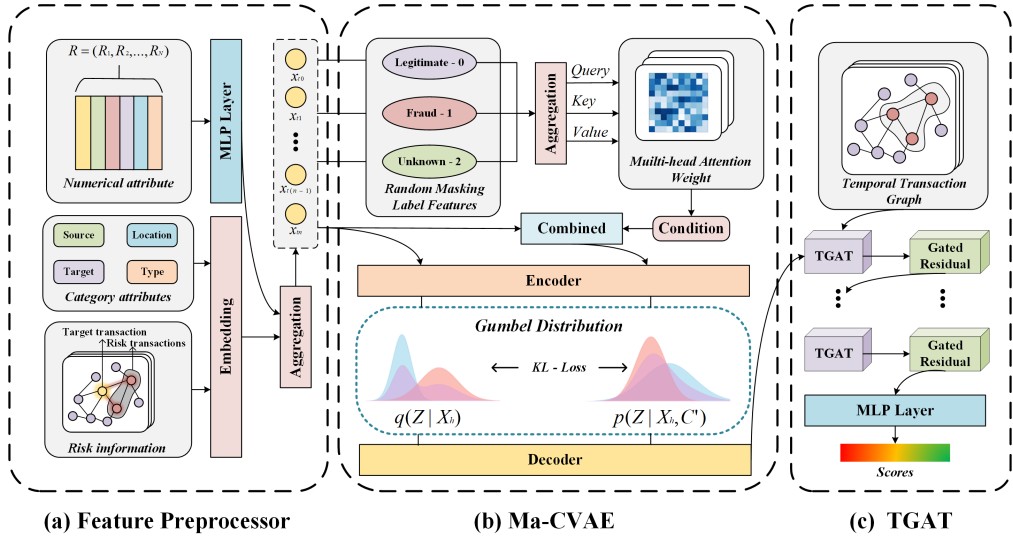

Figure 2: The illustration of the overview of the adaptivae semi-supervied method for CCFD.

capture temporal patterns in transaction sequences. However, traditional machine learning methods rely on manually designed features, limiting their ability to capture complex patterns, while earlier deep learning approaches struggle to integrate both spatial and temporal patterns, hindering their effectiveness in large-scale, real-world fraud detection systems. Graph-based learning models have gained popularity in CCFD, where each transaction is represented as a node with features such as account details and merchants, and the connections between transactions are modeled as edges. Recent works like those of Shi et al. (2022) and Liu et al. (2021a) have shown that GNNs, particularly when using attention and temporal structures, perform well in fraud detection tasks.

## 3 METHODS

### 3.1 MODEL ARCHITECTURE

**Problem Formalization** Given a series of credit card transactions defined as $\mathbf{R} = \{\mathbf{R_1}, \mathbf{R_2}, \ldots, \mathbf{R_n}\}$, each transaction $\mathbf{R_n}$ comprises various attributes $\mathbf{R_n} = \{\mathbf{S}, \mathbf{L}, \mathbf{Ta}, \mathbf{Ty}\}$. Here, $\mathbf{S}$ denotes the source of the transaction initiator, $\mathbf{L}$ indicates the transaction location, $\mathbf{Ta}$ represents the target party receiving the payment, and $\mathbf{Ty}$ refers to the transaction type, such as online shopping or cash withdrawals. The subset $\mathbf{D} \subseteq \mathbf{R}$ consists of target transaction events for fraud detection, aiming to predict the probability of credit card fraud in the target transaction $t_{i+1}$ based on historical records $t_1, \ldots, t_i$.

### 3.1.1 FEATURE PREPROCESSING AND ATTRIBUTE EMBEDDING

Following Xiang et al. (2023), all user transaction records are retained, including those with few authorized transactions, to avoid overlooking potential fraud cases. Time-series features are constructed by sorting all transactions in the dataset $\mathbf{R}$ chronologically by timestamp. For each time window, features are extracted from historical transactions $t_1, \ldots, t_i$ and the target transaction $t_{i+1}$, with the target's label serving as the time window label. Inspired by Fu et al. (2016), metrics such as average, total, standard deviation of transaction amounts, and the difference between current and average amounts are calculated. Following Cheng et al. (2020), the number of transactions, distinct targets (merchants), transaction locations, and types of transactions within the time window are also counted. These computed features, combined with original transaction attributes (source $\mathbf{S}$, location $\mathbf{L}$, target $\mathbf{Ta}$, type $\mathbf{Ty}$), create a comprehensive feature set for the CCFD model, denoted as $\mathbf{U} \in \mathbb{R}^{N \times d}$, where $N$ is the number of transaction records and $d$ is the feature dimension.

To enhance feature representation, following Xiang et al. (2023), a time window $T$ is set to compute transaction features over a specific period. Numerical attributes such as transaction amount and distinct categories associated with the user are calculated within this window, resulting in the eight features listed in Appendix Table 4. These, combined with the original numerical attributes of the target transaction, form the new numerical attribute set $\mathbf{X}_{na}$, providing better temporal correlations and relationships between transactions that isolated features cannot capture.

As shown in Figure 2 (a), the CCFD model consists of feature preprocessing and risk embedding. The Feature Preprocessor extracts three inputs from $\mathbf{U} \in \mathbb{R}^{N \times d}$: Numerical attributes $\mathbf{X}_{na}$, Categorical attributes $\mathbf{X}_{ca}$, and Risk information $\mathbf{X}_r$. Numerical attributes are extracted from $\mathbf{U}$ as $\mathbf{X}_{na} \in \mathbb{R}^{N \times d_{na}}$. Categorical attributes are one-hot encoded to $\mathbf{X}_{ca} \in \mathbb{R}_{ca}^{N \times d}$, with $d_{ca} = d_{na} = d$. For risk information $\mathbf{X}_r$, inspired by Shi et al. (2020), a unified approach for label and feature propagation is proposed, allowing simultaneous updates of node features and labels. Each transaction's manually labeled status ('legitimate' or 'fraud') is treated as a node feature, while 'unlabeled' nodes are assigned zero embeddings, producing $\mathbf{X}_r \in \mathbb{R}^{N \times d_r}$, where $d_r = d$.

The categorical attributes $\mathbf{X}_{ca}$, numerical attributes $\mathbf{X}_{na}$, and risk information $\mathbf{X}_r$ are then combined to form the node feature representation $\mathbf{X}_h \in \mathbb{R}^{N \times d}$, input to Ma-CVAE as shown in Figure 2(b).

$$\mathbf{X}_h = \text{OneHot}(\mathbf{U}_{\text{categorical}}) + f_{\text{mlp}}(\mathbf{U}_{\text{numerical}}) + f_{\text{e}}(\mathbf{Y}_r) \quad = \mathbf{X}_{ca} + \mathbf{X}_{na} + \mathbf{X}_r \qquad (1)$$

### 3.1.2 MULTIHEAD ATTENTION - CONDITIONAL VARIATIONAL AUTOENCODER

To simulate a realistic scenario, the Random Label Masking (RLM) model randomly masks some labels as 'unknown' (value 2), creating semi-supervised data. This mitigates overfitting issues associated with self-loops and GNN-based models Shi et al. (2020); Giraldo et al. (2023); Rusch et al. (2023). The processed labels, represented as $\mathbf{X}_l \in \mathbb{R}^N$, are used as inputs for the Ma-CVAE model, as shown in Figure 2 (b). A label embedding function maps $\mathbf{X}_l$ to the dimension of spatial features, resulting in $\mathbf{X}_l^{'} \in \mathbb{R}^{N \times d}$. However, some legitimate labels are too similar to fraud labels, potentially misleading the learning process. To address this, a Multi-Head Attention mechanism Vaswani et al. (2017) enhances feature representation by focusing on label information with clear distinguishing characteristics.

Then, the Multi-Head Attention mechanism processes the input $\mathbf{X}_l^{'}$ by first adding a dimension, transforming it into $\mathbb{R}^{n \times 1 \times d}$. It is then passed through fully connected layers and multiplied by the weight matrices $\mathbf{W}_q \in \mathbb{R}^{d \times k}$, $\mathbf{W}_k \in \mathbb{R}^{d \times k}$, and $\mathbf{W}_v \in \mathbb{R}^{d \times d_v}$. Scaled dot-product attention scores are calculated and normalized to form the attention matrix, which is then multiplied by $\mathbf{V}_l$ to focus on specific label information. Each attention head outputs a different perspective on the label relationships, preventing bias toward a small subset of features Duan et al. (2022). The attention outputs from multiple heads are concatenated and transformed using a linear layer, $\mathbf{W}_h \in \mathbb{R}^{h d_v \times d}$, represented as $\text{Concat}(\mathbf{x}_1, \mathbf{x}_2, \ldots, \mathbf{x}_{\text{head}})$. The result, $\mathbf{X}_a \in \mathbb{R}^{n \times 1 \times d}$, is then squeezed to remove the second dimension, producing $\mathbf{X}_a^{'} \in \mathbb{R}^{n \times d}$. The final output is calculated as:

$$\mathbf{C} = \text{LN}(\text{squeeze}(\mathbf{W}_h \text{Concat}(\mathbf{x}_1, \mathbf{x}_2, \ldots, \mathbf{x}_{\text{head}}))) + \mathbf{X}_l^{'} \qquad (2)$$

In Eq. 2, $\mathbf{W}_h$ is the transformation matrix, and LN denotes layer normalization. The residual connection between $\mathbf{X}_a^{'}$ and the original input $\mathbf{X}_l^{'}$ improves feature representation. The resulting matrix, $\mathbf{C} \in \mathbb{R}^{n \times d}$, serves as a condition for subsequent Gumbel distribution sampling in the latent space Jang et al. (2017).

Inspired by Sohn et al. (2015), supervised learning often faces challenges in learning complex representations, leading to representation bias. In the CCFD task, capturing diverse fraud features is similarly difficult. To tackle this, features are sampled from a latent space informed by conditional prior information, such as label data, enhancing the model's ability to capture complex structures.

Traditional graph-based models frequently encounter oversmoothing, where node representations converge to indistinguishable values as the number of layers increases. This phenomenon can be formally quantified using a node similarity measure $\mu : \mathbb{R}^{N \times d} \to \mathbb{R}_{\geq 0}$, which evaluates the diversity among node features. Oversmoothing occurs when the node similarity measure converges to zero over time, defined as:

$$\lim_{l \to \infty} \mu(\mathbf{X}^{(l)}) = 0. \qquad (3)$$

In particular, oversmoothing occurs at an exponential rate, where the constant $C_1 > 0$ represents the initial value of the similarity measure at $l = 0$, and $C_2 > 0$ controls the rate of decay. A larger $C_2$

results in faster decay, meaning node features become similar more quickly as the number of layers increases. Specifically, for any $l \in \mathbb{N}$, the following holds:

$$\mu(\mathbf{X}^{(l)}) \leq C_1 e^{-C_2 t}. \tag{4}$$

For our TGAT models, the similarity measure $\mu(\mathbf{X})$ is computed as:

$$\mu(\mathbf{X}) = \|\mathbf{X} - 1\gamma_{\mathbf{X}}\|_F, \tag{5}$$

where $\gamma_{\mathbf{X}} = \frac{1^\top \mathbf{X}}{N}$ represents the average of the node features. As $\mu(\mathbf{X})$ decreases, node representations become increasingly similar, leading to a loss of feature diversity.

To mitigate this issue, the Ma-CVAE model introduces a mechanism that increases feature diversity through a combination of conditional sampling and the Gumbel-Softmax distribution. In Equation 6, the input features $\mathbf{X}_h$ and the condition $\mathbf{C}'$, weighted by Attention, are concatenated and passed through the encoder to obtain $\mathbf{Z}_{\text{logits}}$. This step is crucial for maintaining diversity among node features and reducing the risk of oversmoothing, as the added condition information helps prevent the convergence of node representations to similar values, thereby keeping $\mu(\mathbf{X})$ sufficiently high.

$$\mathbf{Z}_{\text{logits}} = \text{Encoder}(\text{Concat}(\mathbf{X}_h, \mathbf{C}')) \tag{6}$$

In the traditional VAE model, the latent space is often represented by a Gaussian distribution, which may not be suitable for capturing the discrete and complex nature of features in CCFD tasks. This limitation can exacerbate oversmoothing as it leads to poor feature differentiation. To overcome this, the Ma-CVAE employs the Gumbel-Softmax distribution for discrete sampling, as shown in Appendix Figure 4, which helps in maintaining feature diversity by providing a better fit for categorical features. This approach can be mathematically represented as:

$$\mathbf{G}(\mu, \beta) = \frac{1}{\beta} \exp\left( -\frac{x-\mu}{\beta} + \exp\left( -\frac{x-\mu}{\beta} \right) \right) \tag{7}$$

The Gumbel distribution perturbs the logits, enabling a differentiable approximation of categorical variables. This allows for more effective gradient propagation through the sampling process, enhancing the model's ability to learn diverse representations and maintain a higher node similarity measure $\mu(\mathbf{X})$. By using the Gumbel-Max trick, the one-hot encoded variable $\mathbf{Z}$ is obtained as a discrete latent variable, increasing feature diversity and preventing the homogenization of node features. As shown in the first part of Eq. 8:

$$\mathbf{Z} = \text{Onehot}\left( \arg\max_i [\mathbf{g}_i + \log \mathbf{Z}_{\text{logits},i}] \right) \tag{8}$$

where $i$ represents different categories of features. However, the $\arg\max$ function used in Eq. 8 is not differentiable. Inspired by Jang et al. (2017), a softmax function is used to approximate and simplify this process as shown in Eq. 9, resulting in $\mathbf{Z} \in \mathbb{R}^{n \times d}$.

$$\mathbf{Z} = \text{Softmax}\left( \frac{\mathbf{Z}_{\text{logits}} + \mathbf{g}}{\tau} \right) \tag{9}$$

In Eq. 9, $\tau$ serves as a hyperparameter for feature distribution. The temperature parameter $\tau$ controls the smoothness of the distribution and helps maintain a balance between discrete and continuous representations. Choosing a larger $\tau$ value ($\tau \geq 10$) promotes uniform learning across features, further enhancing diversity and preventing the dominance of certain features, thus effectively reducing the oversmoothing effect. The final decoded output, $\mathbf{X}_{\text{cvae}} \in \mathbb{R}^{n \times d}$, combines the resampled result $\mathbf{Z}$ with the condition $\mathbf{C}'$ and passes through a decoder composed of fully connected layers, as shown in the following equation:

$$\mathbf{X}_{\text{cvae}} = \text{Decoder}(\mathbf{Z} \| \mathbf{C}') + \mathbf{X}_h \tag{10}$$

By incorporating the condition $\mathbf{C}'$ as an additional guide for learning, the Ma-CVAE model effectively captures the latent distribution of data points and the unique characteristics of each transaction.

### 3.1.3 GATED TEMPORAL ATTENTION NETWORK

After the Ma-CVAE generates the latent feature representations, temporal information features can be represented as $\mathbf{X}_{\text{temp}} = \{x_{t_0}, x_{t_1}, \ldots, x_{t_n}\}$, where $x_{t_i} = x_n^{t_i} + x_c^{t_i}$. Here, $x_n^{t_i}$ and $x_c^{t_i}$ denote the numerical and categorical features that vary over time, respectively. Setting $\mathbf{h} = \mathbf{X}_{\text{temp}}$, the TGAT

model, integrated with a two-layer MLP as proposed by. Xiang et al. (2023), is employed for risk prediction by learning the temporal features $\mathbf{X}_{\text{temp}}$.

Traditional GCN and GNN models use static graph structures for information aggregation, with GNNs applying uniform weights to all neighbors, risking noise amplification. While GCNs mitigate this by normalizing node contributions, both struggle with dynamic environments like temporal graphs. The TGAT model Xiang et al. (2023), based on GAT, is particularly effective for such graphs. Unlike GCNs that rely on Laplacian transformations, TGAT utilizes can dynamically assign weights to neighboring node features based on their relevance, as illustrated in Figure 6 (b). Each single-head operation in TGAT can be expressed as follows.

A learnable parameter $\mathbf{W}^{(l)} \in \mathbb{R}^{d \times d}$ is used to obtain the hidden layer features through a linear transformation. Then, the attention score $e_{ij}^{(l)}$ between the target node $i$ and its neighbor node $j$ is calculated. This attention score quantifies the weight of information flow from node $j$ to node $i$ and represents the unnormalized attention score for the edge $(i, j)$ in layer $l$. Specifically, the features of the target node and its neighbor, $\mathbf{z}_i^{(l)}$ and $\mathbf{z}_j^{(l)}$, are concatenated. Then, this concatenated feature vector is multiplied by a learnable weight vector $\mathbf{a}^{(l)T}$, and the result is passed through the LeakyReLU activation function to obtain the attention score. Then, the attention score $e_{ij}^{(l)}$ is then normalized to map it to the range $[0, 1]$, ensuring that the sum of attention scores for each target node across its different neighbors equals 1:

$$\alpha_{ij}^{(l)} = \frac{\exp(e_{ij}^{(l)})}{\sum_{k \in \mathcal{N}(i)} \exp(e_{ik}^{(l)})} \tag{11}$$

After obtaining the attention scores, the features of all neighboring nodes of the target node are aggregated together, weighted by their respective attention scores. This aggregated feature vector is then passed through the activation function $\sigma$ to provide a non-linear transformation, generating the feature representation for the next layer:

$$\mathbf{h}_i^{(l+1)} = \sigma \left( \sum_{j \in \mathcal{N}(i)} \alpha_{ij}^{(l)} \mathbf{z}_j^{(l)} \right) \tag{12}$$

The principle diagram of TGAT is shown in Appendix Figure 6.

### 3.1.4 Gated Residual

To maintain a balance between preserving original information and introducing new information, and to prevent either from excessively influencing the model's training, which could lead to increased training error as the network depth increases, an Attribute-driven Gated Residual approach, as proposed by Xiang et al. (2023), is employed. This approach provides a shorter backpropagation path, allowing gradients to flow more easily to the shallower layers, effectively mitigating the problem of gradient vanishing or explosion.

A learnable parameter $\mathbf{W}_{\text{skip}}$ is used to perform a linear transformation on the feature $\mathbf{h}_i^{(l)}$ of node $i$ at layer $l$, yielding the skip feature $\mathbf{h}_{\text{skip}}$. This allows the feature from the previous layer to be introduced into the current layer:

$$\mathbf{h}_{\text{skip}} = \mathbf{W}_{\text{skip}} \mathbf{h}_i^{(l)} \tag{13}$$

Subsequently, a gating value $\mathbf{g}_i$ is introduced to determine the fusion ratio between the skip feature $\mathbf{h}_{\text{skip}}$ and the new feature $\mathbf{h}_i^{(l+1)}$. Specifically, the skip feature, the new feature, and their difference are first concatenated, then passed through a learnable weight matrix $\mathbf{W}_{\text{gate}}$ for a linear transformation, and finally processed by an activation function $\sigma$ to obtain the gating value $\mathbf{g}_i$. The gating value ranges between $[0, 1]$ and is used to control the degree of mixing between the original information and the new information in the final output. The formula is as follows:

$$\mathbf{g}_i = \sigma \left( \mathbf{W}_{\text{gate}} \left[ \mathbf{h}_{\text{skip}} \parallel \mathbf{h}_i^{(l+1)} \parallel \left( \mathbf{h}_{\text{skip}} - \mathbf{h}_i^{(l+1)} \right) \right] \right) \tag{14}$$

where $\parallel$ denotes the concatenation of the input vectors. The output feature representation $\mathbf{h}_i^{(l+1)}$ is then computed as a weighted sum of the skip feature and the newly aggregated feature, controlled

by the gating value:

$$\mathbf{h}_i^{(l+1)} = \mathbf{g}_i \odot \mathbf{h}_{\text{skip}} + (1 - \mathbf{g}_i) \odot \mathbf{h}_i^{(l+1)} \qquad (15)$$

where $\odot$ denotes element-wise multiplication. The Gated Residual, instead of a simple concatenated residual, allows the model to adaptively balance original and transformed features. Then, to accelerate convergence, layer normalization and an activation function are then applied to the output features.

$$\mathbf{X}_{\text{out}} = \sigma(\text{LayerNorm}(\mathbf{h}_{\text{out}})) \qquad (16)$$

where $\mathbf{X}_{\text{out}}$ is the final output representation of the TGAT model, $\sigma$ denotes the activation function, and LayerNorm represents the layer normalization operation applied to the input feature $\mathbf{h}_{\text{out}}$.

### 3.1.5 LOSS FUNCTIONS AND OPTIMIZATION

**Fraud Risk Prediction** After obtaining the output $\mathbf{X}_{\text{out}}$ from TGAT Xiang et al. (2023), it is used in a multi-layer MLP to calculate the probability of a fraud label in the CCFD task. The probability is computed as $\hat{\mathbf{r}} = \sigma(\alpha(\mathbf{X}_{\text{out}}\mathbf{W}_1 + \mathbf{c}_1)\mathbf{W}_2 + \mathbf{c}_2)$, where $\sigma$ is the activation function mapping the probability to the range [0,1]. A cross-entropy loss is then used to measure the quality of the model predictions, resulting in the loss function for each training iteration, which is used for backpropagation:

$$\mathcal{L}_1 = -\frac{1}{M} \sum_{j=0}^{M} \left[ z_j \cdot \log q(\hat{r}_j | \mathbf{Y}, \mathbf{A}) + (1 - z_j) \cdot \log(1 - q(\hat{r}_j | \mathbf{Y}, \mathbf{A})) \right], \qquad (17)$$

where $\mathbf{A}$ represents the prior information input to the model, $z$ denotes the target label of transactions, and $M$ represents the total number of transactions.

**Reconstructed and KL Loss** To prevent excessive dispersion in VAE-based models from causing reconstruction distortion, Kingma & Welling (2013) introduced reconstruction loss to constrain the model during backpropagation. Similarly, in the Ma-CVAE model, a reconstruction loss $\mathcal{L}_{\text{re}} = \text{MSE}(\mathbf{X}'_{\mathbf{h}}, \mathbf{X}_{\mathbf{h}})$ and a KL divergence loss $\mathcal{L}_{\text{KL}} = D_{\text{KL}}(q(\mathbf{Z}|\mathbf{X}_{\mathbf{h}}, \mathbf{C}') \| p(\mathbf{Z}|\mathbf{C}'))$ are combined to form the total loss, weighted by a factor $\alpha$. Finally, the model's fraud risk prediction loss $\mathcal{L}_1$ and the Ma-CVAE's loss $\mathcal{L}_2 = \mathcal{L}_{\text{re}} + \alpha\mathcal{L}_{\text{KL}}$ are weighted by a factor $\beta$ and summed together for parameter updates. The total loss is expressed as:

$$\mathcal{L}_{\text{total}} = \mathcal{L}_1 + \beta\mathcal{L}_2 = \mathcal{L}_1 + \beta(\mathcal{L}_{\text{re}} + \alpha\mathcal{L}_{\text{KL}}) \qquad (18)$$

## 4 EXPERIENCES

### 4.1 EXPERIMENT SETTINGS

The proposed model is evaluated on the Financial Fraud Semi-supervised Dataset (FFSD) Xiang et al. (2023) and two review fraud detection datasets: the Amazon review dataset Jha et al. (2018) and the YelpChi review dataset Rayana & Akoglu (2015), as shown in Appendix Figure 5. The **FFSD** contains 1,820,840 transaction records (90.35% unlabeled, 7.79% legitimate, 1.86% fraudulent) and 31,619,440 connections. The **Amazon dataset** consists of 11,948 review nodes and 8,808,728 connections (93.13% legitimate, 6.87% fraudulent), focusing on fraudulent reviews in the Musical Instrument category. The **YelpChi dataset** includes 45,954 review nodes and 3,846,979 connections (85.47% legitimate, 14.53% fraudulent), targeting hotel and restaurant reviews. Our model is compared with 9 SOTA fraud detection models: GEM Liu et al. (2018), Player2Vec Wang et al. (2019c), FdGars Zhang et al. (2019), Semi-GNN Wang et al. (2019a), GraphSAGE Hamilton et al. (2017), GraphConsis Liu et al. (2020), CARE-GNN Dou et al. (2020), PC-GNN Liu et al. (2021b), and TGAT Xiang et al. (2023). Performance on CCFD and opinion fraud detection datasets is evaluated using AUC, F1-macro, and AP, which assess precision-recall balance and classification capability based on True Positives (TP), False Positives (FP), and False Negatives (FN).

### 4.2 DETECTION PERFORMANCE ACROSS DATASETS

To comprehensively evaluate the performance of various models across different datasets, Tables 1 and 4.2 summarize the AUC, F1, and AP metrics for the YelpChi, Amazon, and FFSD datasets. The results demonstrate that the Ma-CVAE+TGAT model consistently outperforms other state-of-the-art

methods. However, the publicly available FFSD is a simulated version of the complete datasets, specifically mentioned in TGAT Xiang et al. (2023). Consequently, the simulated FFSD dataset serves as a baseline for testing in Table 1, with entries marked with * indicating results obtained using this dataset. For further comparison with previous methods, results on the simulated FFSD dataset are presented in the appendix (Table A).

The Ma-CVAE+TGAT model exhibited outstanding performance across multiple datasets. On the simulated **FFSD** dataset, it achieved an AUC of 0.8406, an F1 score of 0.7362, and an AP of 0.7104, surpassing the previous SOTA method, TGAT*, which recorded an AUC of 0.8286, an F1 of 0.7336, and an AP of 0.6585. This represents improvements of 1.45% in AUC, 0.35% in F1, and 7.86% in AP, highlighting Ma-CVAE+TGAT's strong capabilities. Although the complete FFSD dataset is not publicly available, the robust performance of TGAT on the full dataset suggests that Ma-CVAE+TGAT would also excel in that context. On the **YelpChi** dataset, the model reached an AUC of 0.9486, an F1 of 0.8446, and an AP of 0.8192, outperforming other models such as PC-GNN and CARE-GNN, with notable improvements of 3.05% in AUC, 3.82% in F1, and 7.88% in AP compared to the best baseline, TGAT. Additionally, for the **Amazon** dataset, Ma-CVAE+TGAT recorded an AUC of 0.9713, an F1 of 0.9242, and an AP of 0.8970, exceeding TGAT by 0.86% in AUC, 0.31% in F1, and 1.49% in AP. These substantial gains, particularly in the AP metric, emphasize the model's effectiveness in scenarios with fewer anomalies and lower anomaly rates, solidifying its position as a leading method in fraud detection tasks.

| Dataset | FFSD | | |
|---|---|---|---|
| | AUC | F1 | AP |
| GEM | 0.5383 | 0.1490 | 0.1889 |
| Player2Vec | 0.5278 | 0.2147 | 0.2041 |
| FdGars | 0.6965 | 0.4089 | 0.2449 |
| Semi-GNN | 0.5473 | 0.4485 | 0.2758 |
| GraphSAGE | 0.6527 | 0.5370 | 0.3844 |
| GraphConsis | 0.6579 | 0.5466 | 0.3876 |
| CARE-GNN | 0.6623 | 0.5771 | 0.4060 |
| PC-GNN | 0.6795 | 0.6077 | 0.4487 |
| TGAT | 0.7616 | 0.6764 | 0.5767 |
| TGAT* | 0.8286 | 0.7336 | 0.6585 |
| Ma-CVAE + TGAT* | **0.8406** | **0.7362** | **0.7104** |

Table 1: Comparison of the performance on FFSD dataset.

| Dataset | YelpChi | | | Amazon | | |
|---|---|---|---|---|---|---|
| | AUC | F1 | AP | AUC | F1 | AP |
| GEM | 0.5270 | 0.1060 | 0.1807 | 0.5261 | 0.0941 | 0.1159 |
| Player2Vec | 0.7003 | 0.4121 | 0.2473 | 0.6185 | 0.2451 | 0.1291 |
| FdGars | 0.7332 | 0.4420 | 0.2709 | 0.6556 | 0.2713 | 0.1438 |
| Semi-GNN | 0.5161 | 0.1023 | 0.1811 | 0.7063 | 0.5492 | 0.2254 |
| GraphSAGE | 0.5364 | 0.4508 | 0.1712 | 0.7502 | 0.5795 | 0.2624 |
| GraphConsis | 0.7060 | 0.6041 | 0.3331 | 0.8782 | 0.7819 | 0.7336 |
| CARE-GNN | 0.7934 | 0.6493 | 0.4268 | 0.9115 | 0.8531 | 0.8219 |
| PC-GNN | 0.8174 | 0.6682 | 0.4810 | 0.9581 | 0.9153 | 0.8549 |
| TGAT | 0.9241 | 0.7988 | 0.7513 | 0.9630 | 0.9213 | 0.8838 |
| Ma-CVAE+TGAT | **0.9486** | **0.8446** | **0.8192** | **0.9713** | **0.9242** | **0.8970** |

Table 2: Comparison of performance on YelpChi and Amazon Datasets.

### 4.3 Impact of the Number of Layers

In Figure 3, the performance of Ma-CVAE + TGAT and TGAT models with varying numbers of layers (L = 1 to 5) on the FFSD dataset is compared to analyze the impact of model depth on capturing temporal patterns. As the number of layers increases from 1 to 4, both models show improvements, with Ma-CVAE + TGAT achieving a peak AUC of 0.8406, F1 of 0.7362, and AP of 0.7104 at $L = 4$, representing increases of 5.94%, 2.42%, and 15.00% respectively compared to $L = 1$. This demonstrates the model's enhanced ability to capture temporal dependencies with additional layers. However, performance declines when the number of layers increases to $L = 5$, indicating the onset of over-smoothing, where node features become indistinguishable. The Ma-CVAE model mitigates this issue by using Gumbel distribution for feature sampling, maintaining distinct features and supporting deeper models. This is evident from the better performance of Ma-

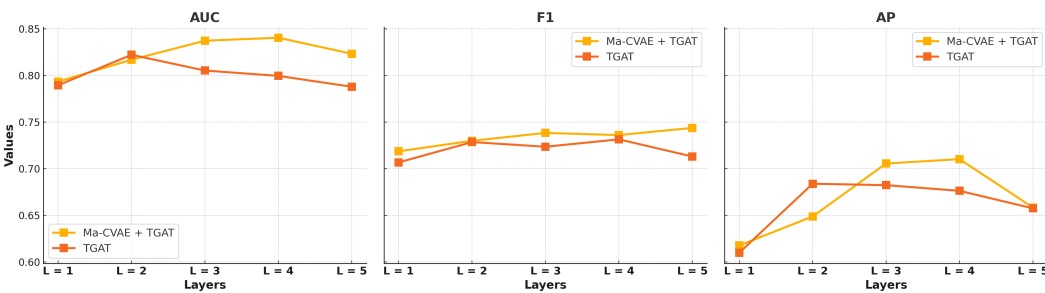

Figure 3: Comparison of Ma-CVAE + TGAT and TGAT model performance across different layers (L = 1 to L = 5) on the FSSD dataset.

CVAE + TGAT at $L = 4$ compared to TGAT alone at $L = 2$, highlighting its capacity to utilize deeper layers effectively without significant performance degradation.

## 4.4 ABLATION STUDY

This subsection aims to examine: 1) the effectiveness of the MA-CVAE module, 2) the impact of various feature preprocessing methods, and 3) the influence of different distribution techniques. [1]

| Dataset | FSSD | | | Amazon | | | YelpChi | | |
|---|---|---|---|---|---|---|---|---|---|
| | AUC | F1 | AP | AUC | F1 | AP | AUC | F1 | AP |
| w/o | 0.8223 | 0.7287 | 0.6840 | 0.9520 | 0.9150 | 0.8604 | 0.8920 | 0.7405 | 0.6601 |
| w | **0.8406** | **0.7362** | **0.7104** | **0.9713** | **0.9242** | **0.8970** | **0.9486** | **0.8446** | **0.8192** |

Table 3: Comparison of the performance across FSSD, Amazon, and YelpChi datasets without Ma-CVAE (w/o) and with Ma-CVAE (w).

To evaluate the contribution of Ma-CVAE, ablation experiments were conducted on the FSSD, Amazon, and YelpChi datasets. As shown in Table 3, the inclusion of Ma-CVAE (w) significantly improved performance across all three datasets compared to the version without it (w/o). Specifically, on the FSSD dataset, the AUC increased from 0.8223 to 0.8406, F1 from 0.7287 to 0.7362, and AP from 0.6840 to 0.7104, representing relative improvements of 2.23%, 1.03%, and 3.86%, respectively. In the Amazon dataset, Ma-CVAE boosted the AUC from 0.9520 to 0.9713 and AP from 0.8604 to 0.8970, reflecting gains of 2.03% and 4.25%. The slight decrease in F1 from 0.9150 to 0.9242 can be attributed to the already high baseline value, making further improvements challenging. The most notable enhancements were observed on the YelpChi dataset, where the AUC increased from 0.8920 to 0.9486 (6.35%), F1 from 0.7405 to 0.8446 (14.06%), and AP from 0.6601 to 0.8192 (24.09%). These results underscore the crucial role of Ma-CVAE in capturing complex feature distributions and effectively handling imbalanced datasets, particularly in challenging classification scenarios where accurate representation of minority classes is essential.

## 5 CONCLUSION

This paper presents the Ma-CVAE+TGAT model for credit card fraud detection, integrating multi-head attention and variational autoencoding to effectively capture diverse transactional features and mitigate the over-smoothing problem inherent in GNN-based models. Experimental results on the FFSD, Amazon, and YelpChi datasets demonstrate the model's superior performance, achieving notable improvements in AUC, F1, and AP metrics over existing state-of-the-art methods. Despite these advances, there are still areas for future improvement. Future work could explore incorporating real-time adaptation mechanisms to address the dynamic nature of fraud patterns and further enhance model robustness.

---

[1]Due to space constraints, a detailed analysis of the impact of feature preprocessing methods and distribution techniques can be found in the Appendix.

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

# A APPENDIX

| Feature Types | Description |
|---|---|
| Trans amount avg | Average amount of the transactions |
| Trans amount total | Total amount of the transactions |
| Trans amount stddev | Standard deviation of the transaction amounts during the past period |
| Trans amount bias | The difference between the amount of this transaction and the average |
| Trans count | Total number of the transactions |
| Trans target count | Number of unique target accounts involved in transactions |
| Trans location count | Number of unique transaction locations |
| Trans type count | Number of unique transaction types |

Table 4: Given a time window $T$, the data for each transaction and account type is aggregated and further processed to derive eight new features. The table includes descriptions of each feature type for CCFD tasks.

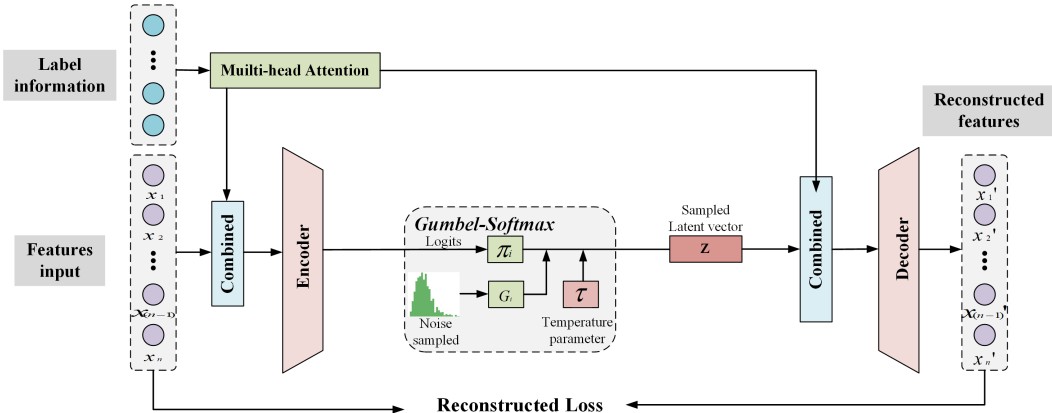

Figure 4: An illustration of the Ma-CVAE model, showcasing the use of the Gumbel-Softmax distribution Jang et al. (2017) to map data through a discrete latent space.

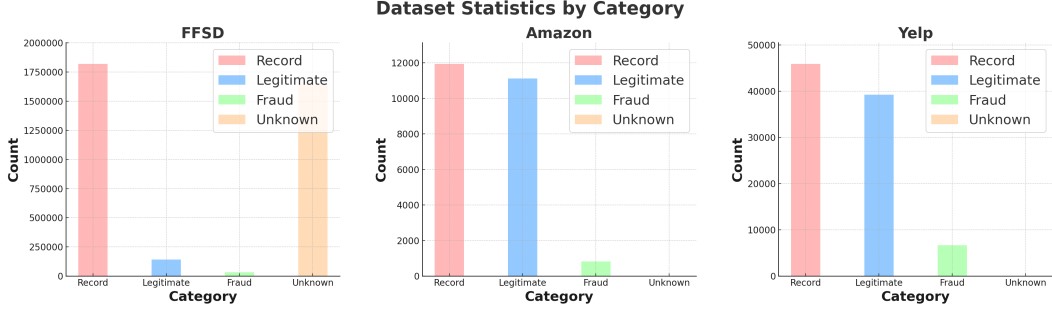

Figure 5: The illustration of the FFSD, Amazon, and Yelp datasets, where red represents records, blue represents legitimate records, green represents fraudulent records, and orange represents un-labelled records. These different categories are manually labeled. Due to the significantly larger number of records in the FFSD dataset compared to Amazon and Yelp, the cost of manual labeling increases substantially, resulting in a large number of unlabelled records.

## A.1 PERFORMANCE COMPARISON OF FEATURE PROCESSING MODELS

As highlighted in the Introduction, GNN-based models with an increasing number of layers can suffer from feature homogenization, which negatively impacts risk prediction accuracy. To address this

| Dataset | FSSD | | |
|---|---|---|---|
| | AUC | F1 | AP |
| MCNN* | 0.7414 | 0.6285 | 0.3126 |
| STAN* | 0.7468 | 0.6399 | 0.3201 |
| STAGN* | 0.7008 | 0.5437 | 0.2659 |
| GADGNN* | 0.7746 | 0.7136 | 0.5981 |
| TGAT* | 0.8286 | 0.7336 | 0.6585 |
| Ma-CVAE + TGAT* | **0.8406** | **0.7362** | **0.7104** |

Table 5: Comparison of the performance on the simulated versions of the FFSD dataset.

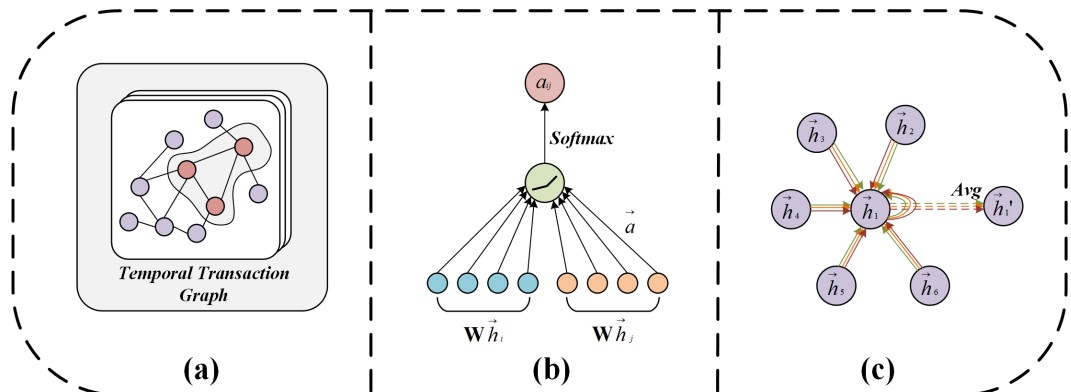

Figure 6: This illustration demonstrates the working principle of the TGAT model, which applies a Temporal Transaction Graph based on the Graph Attention Networks (GAT) Veličković et al. (2018). The figure is composed of three subfigures: (a) shows the Temporal Transaction Graph obtained through data processing; (b) illustrates how each node in the Temporal Transaction Graph calculates the Attention by applying learned weight parameters to the features of both its neighbors and itself, followed by a softmax activation function; (c) depicts the multi-head attention mechanism, where each color represents a different attention head. Compared to a single attention mechanism, the multi-head attention mechanism is better at learning the latent representations of features.

issue, the proposed Ma-CVAE model was compared with other mainstream generative and feature processing models, including the Variational AutoEncoder (VAE) Kingma & Welling (2013), Conditional Diffusion Zhang et al. (2023), AutoEncoder (AE) Hinton & Salakhutdinov (2006), and Convolutional AutoEncoder (ConvAE) Thill et al. (2021). The goal was to alleviate the over-smoothing problem present in the TGAT model. The results of these experiments, conducted on three datasets (FSSD, Amazon, and YelpChi), are summarized in Table 6. The evaluation metrics used were AUC, F1, and AP, with the best values highlighted in bold.

The Ma-CVAE model demonstrated superior performance across all three datasets. On the FSSD dataset, Ma-CVAE achieved an AUC score of 0.8406, outperforming the AE Hinton & Salakhutdinov (2006) model by 0.1040, representing an improvement of 14.12%. This indicates that Ma-CVAE effectively captures the differences between positive and negative samples. In terms of F1 and AP, Ma-CVAE also showed significant enhancements, reaching scores of 0.7362 and 0.7104, respectively, which correspond to improvements of 63.08% and 45.70% over the AE model. These results emphasize Ma-CVAE's capability in handling imbalanced datasets effectively.

For the Amazon dataset, the Ma-CVAE model achieved an AUC score of 0.9713, surpassing the AE model by 0.0513. This result underscores Ma-CVAE's ability to adaptively focus on important features through its multi-head attention mechanism, particularly in high-risk transactions. Additionally, Ma-CVAE outperformed other models in terms of F1 and AP, achieving scores of 0.9142 and 0.8970, respectively, indicating its robustness in feature extraction under complex scenarios.

| Dataset | FSSD | | | Amazon | | | YelpChi | | |
|---|---|---|---|---|---|---|---|---|---|
| | AUC | F1 | AP | AUC | F1 | AP | AUC | F1 | AP |
| AE | 0.7366 | 0.4513 | 0.4876 | 0.9200 | 0.9091 | 0.8081 | 0.8823 | 0.7429 | 0.6472 |
| ConvAE | 0.7469 | 0.4513 | 0.5005 | 0.9559 | 0.9189 | 0.8682 | 0.8701 | 0.7034 | 0.6120 |
| VAE | 0.8069 | 0.7325 | 0.6357 | 0.9592 | 0.9206 | 0.8724 | 0.8687 | 0.7174 | 0.6048 |
| Con-Diffusion | 0.7929 | 0.7277 | 0.6268 | 0.9559 | 0.9189 | 0.8682 | 0.8765 | 0.7264 | 0.6244 |
| **Ma-CVAE** | **0.8406** | **0.7362** | **0.7104** | **0.9713** | **0.9142** | **0.8970** | **0.9486** | **0.8446** | **0.8192** |

Table 6: Comparison of the proposed Ma-CVAE model with other feature processing models, such as the generative model VAE Kingma & Welling (2013) and the diffusion model Con-Diffusion Zhang et al. (2023), across three datasets: FSSD, Amazon, and YelpChi. The evaluation metrics used are AUC, F1, and AP, with the best values highlighted in bold.

On the YelpChi dataset, the Ma-CVAE model maintained its leading position with an AUC score of 0.9486, representing a 7.52% increase over the AE model. The F1 and AP scores also exhibited substantial gains, reaching 0.8446 and 0.8192, respectively. This demonstrates the model's adaptability and robustness across different datasets, effectively handling a variety of data distributions.

The experiments further explored the impact of Conditional supervision on model performance. It was found that incorporating label information into the learning process led to modest performance gains for models like AE and ConvAE. However, the proposed Ma-CVAE model went a step further by integrating a multi-head attention mechanism, which selectively emphasizes high-risk transaction labels and their associated features, resulting in significant improvements across all metrics. To overcome the limitations of traditional VAE models in fitting discrete data, the Ma-CVAE model employed the Gumbel-softmax sampling technique Jang et al. (2017). This method mitigates the challenges associated with Gaussian distributions in handling categorical data, resulting in more stable gradient propagation and higher-quality generated data. Compared to the VAE model, Ma-CVAE achieved an AUC improvement of 0.0337 on the FSSD dataset, and 0.0121 and 0.0799 on the Amazon and YelpChi datasets, respectively. Although diffusion models like Con-Diffusion Zhang et al. (2023) showed some potential, their performance was lower than that of the VAE model, especially on the FSSD dataset, where an AUC of only 0.7929 was achieved. This performance lag might be due to the intrinsic challenges of modeling diverse data distributions using noisy data. The results indicate that the Ma-CVAE model, with its adaptive attention mechanism and effective handling of discrete features, provides a significant performance boost over existing models, particularly in complex, real-world scenarios.

## B COMPARISON OF DISTRIBUTION METHODS IN MA-CVAE

| Dataset | YelpChi | | |
|---|---|---|---|
| | AUC | F1 | AP |
| No Ma-CVAE | 0.8920 | 0.7405 | 0.6601 |
| Ma-CVAE(Gaussian) | 0.9448 | 0.7739 | 0.7670 |
| Ma-CVAE(Gumbel) | **0.9486** | **0.8446** | **0.8192** |

Table 7: Comparison of the performance on the YelpChi dataset without Ma-CVAE and with Ma-CVAE using two different sampling methods: Gumbel Jang et al. (2017) and Gaussian distributions.

To effectively learn the latent representations of features, Ma-CVAE utilizes a method that encodes these features into a latent space, applying a specific distribution to model them Rezende et al. (2014); Kingma & Welling (2013); Kingma et al. (2015). The traditional VAE, which relies on Gaussian distribution for resampling, often struggles with discrete data. As shown in Figure 2, categorical and numerical attributes are combined and input into the model. When these features exhibit discrete characteristics, forcing them to conform to a Gaussian distribution can lead to mismatches in the latent variable distribution, thereby affecting the training process.

In experiments on the FFSD, Amazon, and YelpChi datasets, using Gaussian distribution sampling proved challenging, particularly for the FFSD and Amazon datasets, due to their discrete feature characteristics. As illustrated in Figure 5, the FFSD and Amazon datasets exhibit significantly greater variability in their features compared to the YelpChi dataset. Inspired by Jang et al. (2017), the Gumbel-Softmax technique was adopted for these datasets to better handle the discrete features during the sampling process.

On the YelpChi dataset, which exhibits more continuous characteristics, the Ma-CVAE(Gaussian) method achieved slightly lower performance compared to the Ma-CVAE(Gumbel) method, with a 0.38% drop in AUC. However, this performance gap was more pronounced in the F1 and AP metrics, where the Gumbel method outperformed by 7.06% and 5.22%, respectively, as shown in Table 7. This indicates that the Gumbel-Softmax distribution is better suited for handling features with discrete and mixed characteristics, even on a dataset that contains both discrete and continuous features. For the FFSD and Amazon datasets, significant gradient propagation issues were encountered with the Ma-CVAE(Gaussian) approach. To ensure a consistent comparison across all datasets and to mitigate these issues, the Ma-CVAE(Gumbel) method was used in all reported experiments in Tables 1 and 4.2. This choice was made to maintain consistency and comparability with other state-of-the-art methods. the Ma-CVAE(Gumbel) method demonstrates robust performance improvements over both the baseline model without Ma-CVAE and the Ma-CVAE(Gaussian) method, particularly on datasets with mixed feature distributions, confirming its effectiveness in handling complex data characteristics.

