# OpenReview forum: "Conditionally Adaptive Graph Attention Networks for Credit Card Fraud Detection"
_ICLR.cc/2025/Conference — ICLR 2025 Conference Withdrawn Submission_

### Official Review · Reviewer_2xjo · 2024-10-18

**Soundness:** 2
**Presentation:** 2
**Contribution:** 1
**Rating:** 3
**Confidence:** 4

**Summary:**

The paper proposes a credit card fraud detection method, based on multi-head attention conditional variational autoencoder and temporal graph attention networks. Compared to some fraud detection baselines, This method achieves the best performance on three datasets.

**Strengths:**

This method solves the oversmoothing problem in the fraud detection task by using multi-head attention conditional variational autoencoder to construct more diverse reconstructed features.

Experiments were conducted on three datasets, which are somewhat convincing.

**Weaknesses:**

I strongly recommend the author to check the expression norms throughout, especially for the figure references. (eg. P1L52 has question marks in the figure citation. And Figure (a) as it appears in P1L54 should be Figure 1(a).)

I think the author needs to dedicate more pages to the motivation of the method. It is not intuitive that addressing oversmoothing of features is reasonable for the fraud detection problem. To summarise, oversmoothing is a broad problem on graph neural networks rather than one specific to fraud detection. So, I don't think this innovativeness will convince the readers.

In addition, I would also suggest that the authors add some recent baselines on fraud detection (such as BWGNN[1] GHRN [2]).

[1] Tang J, Li J, Gao Z, et al. Rethinking graph neural networks for anomaly detection[C]//International Conference on Machine Learning. PMLR, 2022: 21076-21089.
[2] Gao Y, Wang X, He X, et al. Addressing heterophily in graph anomaly detection: A perspective of graph spectrum[C]//Proceedings of the ACM Web Conference 2023. 2023: 1528-1538.

**Questions:**

Please see the Weakness part. Thank you.

---

### Official Review · Reviewer_rK1w · 2024-10-31

**Soundness:** 2
**Presentation:** 3
**Contribution:** 2
**Rating:** 5
**Confidence:** 4

**Summary:**

This paper proposes a Multi-head Attention-Conditional Variational Autoencoder model that integrates numerical, categorical, and transaction-related features. It enhances feature diversity during training and improves sensitivity to variations by mapping features into the Gumbel-Softmax space, effectively mitigating over-smoothing issues in graph-based methods. They apply a multi-head attention mechanism to label information, the method better handles label distributions, maximizing the benefits of semi-supervised learning.
In experiments, the model surpasses state-of-the-art methods across multiple metrics in fraud detection tasks.

**Strengths:**

1. The presentation and figures are clear and the methodology is eary to follow.
2. Experimental results on the FFSD, Amazon, and YelpChi datasets demonstrate the model’s superior performance.

**Weaknesses:**

1. There are some formatting issues, such as on line 051, where it reads "Figure ??(a)."

2. The primary focus of this research is on CCFD, but it is crucial to address why the transaction is detected as fraudulent. Providing the corresponding rationale is essential for governmental supervision; otherwise, regardless of the percentage gain in open benchmarks, the findings cannot be applied in real-world scenarios.

3. The technical contribution is limited; it combines TGAT and multi-head attention using Ma-CVAE, while the key idea of mapping features into the Gumbel-Softmax space has been proposed in previous work.

4. The reproducibility of the results is weak, as there are no implementation details or open-source code provided.

5. The paper emphasizes that the model can resolve the over-smoothing issue, but it lacks specific experiments to verify this claim.

6. Another concern is that the real transaction network is extremely large, raising questions about the efficiency of this model. Given that the multi-head attention mechanism and TGAT are computationally expensive, an analysis of their efficiency is necessary.

**Questions:**

1. What are the major challenges in combining TGAT and multi-head attention using Ma-CVAE, and what is the technical novelty of this work?

2. It seems odd that in the ablation study, removing the MA-CVAE module results in worse model performance in Table 3 compared to TGAT in Table 2. What accounts for this phenomenon?

3. The performance gain compared to the baseline is limited on the Amazon dataset. Why is that? Are there any factors related to the dataset that influence the performance of MA-CVAE?

4. Where are the implementation details for the hyperparameters alpha and beta in the loss function? What impact do these two factors have when adjusted? Some explanations or sensitivity experiments are needed.

5. In the real world, fraudulent transactions may account for less than 0.01% of all transactions. How does this model address such a situation?

---

### Official Review · Reviewer_tt96 · 2024-11-01

**Soundness:** 3
**Presentation:** 3
**Contribution:** 3
**Rating:** 8
**Confidence:** 4

**Summary:**

This paper proposes a graph attention network model integrated with multi-head attention and conditional variational autoencoders (Ma-CVAE) for credit card fraud detection tasks. The model effectively addresses the oversmoothing issue in graph neural networks by enhancing feature diversity in the Gumbel-Softmax space and applies a multi-head attention mechanism to label information, leveraging semi-supervised learning to improve handling of imbalanced datasets. As a result, it achieves superior performance over existing state-of-the-art methods on multiple datasets.

**Strengths:**

- The paper innovatively addresses the oversmoothing problem in GNN-based methods for the CCFD task by proposing a Gumbel-softmax feature mapping and conditional sampling.
- The paper provides clear explanations of the model's details, with illustrations that greatly aid in understanding the model's implementation.
- Experimental results demonstrate the capability of the proposed modules in enhancing detection accuracy and alleviating the oversmoothing problem，which is an important breakthrough.

**Weaknesses:**

- The logic in the experiment analysis section of the paper is somewhat weak. For example, in section 4.2, the paper states that the robust performance of TGAT on the full dataset suggests that Ma-CVAE+TGAT would also excel in that context. However, since TGAT* is better than TGAT, the results on the simulated dataset may be overestimated, implying that Ma-CVAE+TGAT may not necessarily outperform TGAT on the complete dataset.
- Some statements in the paper are not rigorous enough. For instance, in section 4.3, it is mentioned that the performance of both models improves as the number of layers increases from 1 to 4, but the performance of TGAT begins to decline after L=2.
- The ablation study on the impact of various feature preprocessing techniques is not rigorous, lacking an introduction to the comparison models and a description of how these models participate in the CCFD task, i.e., experimental details and controlled variables. Moreover, the section also mentions the impact of conditional supervision on model performance, stating that "incorporating label information into the learning process leads to modest performance gains for models like AE and ConvAE," but the experimental results are not shown.
- The experimental analysis in the paper is dull and formulaic. In Appendix A.1, the paper explores the impact of various feature preprocessing methods, analyzing the results on three datasets one by one, which seems unnecessary as the analysis is quite similar across datasets. The analysis of the experimental results in Table 6 is also thin, with the proposed model compared only with the AE model.
- There is a lack of ablation studies to individually verify the effects of conditional sampling.

**Questions:**

- Why is the accuracy of STAGN so poor in Table 5 for the simulated FFSD dataset? This is significantly different from the results in the paper that proposed this model and the results from the model's open-source code.
- The term "FSSD" appears in many places throughout the paper, which should be "FFSD", right?
- What is $\mathbf Y$ in the cross-entropy loss formula in section 3.1.5? It has not been mentioned before. What is its relationship with $\mathbf X_{\text{out}}$?
- In the experimental section, using TGAT to refer to the model proposed by T Xiang et al. (2023) is not complete. The model they proposed should be called GTAN, and TGAT is only a part of GTAN. The model used for the experiment should be the complete GTAN model, right?

---

### Official Review · Reviewer_EGHr · 2024-11-04

**Soundness:** 3
**Presentation:** 3
**Contribution:** 2
**Rating:** 3
**Confidence:** 4

**Summary:**

This paper proposes a GNN-based method for credit card fraud detection. The main focus is to alleviate the oversmoothing issue in GNN as the graph layers increase. For this purpose, a multi-head attention conditional variational autoencoder method is proposed, which can leverage the weight distributions and the Gumbel softmax distribution to reduce the feature homogenization. Experiments on three datasets demonstrates that the proposal can achieve the SOTA performance.

**Strengths:**

(1) The Ma-CVAE model maps the features into the Gumber-Softmax space to make the feature more distinguishing, which is sound and  validated that it can enhance the feature diversity.
(2) The paper is generally well-written and extensive experiments are conducted to evaluate the effectiveness.

**Weaknesses:**

(1) The oversmoothing issue of GNN is not a new problem, and various approaches have been proposed. The authors didn’t compare the proposal with previous oversmoothing mitigation methods, and thus the novelty of the proposal is unclear.  The existing ovresmoothing mitigation works can refer to the section of related works in “Multi-Track Message Passing: Tackling Oversmoothing and Oversquashing in Graph Learning via Preventing Heterophily Mixing. ICML 2024”.
(2) The related works and references on other GNN oversmoothing mitigation methods are missing.

**Questions:**

I'm wondering the novelty of the proposed technique to mitigate the oversmoothing issue in GNN and how about the performance when it is compared with the other oversmoothing alleviation methods.

---

### Note · Authors · 2024-11-20

I have read and agree with the venue's withdrawal policy on behalf of myself and my co-authors.